# Co-Expression of *ZmVPP1* with *ZmNAC111* Confers Robust Drought Resistance in Maize

**DOI:** 10.3390/genes14010008

**Published:** 2022-12-20

**Authors:** Shengxue Liu, Xiaohu Liu, Xiaomin Zhang, Shujie Chang, Chao Ma, Feng Qin

**Affiliations:** 1College of Biological Sciences, China Agricultural University, Beijing 100193, China; 2State Key Laboratory of Crop Stress Adaptation and Improvement, School of Life Sciences, Henan University, Kaifeng 475004, China

**Keywords:** maize, drought resistance, gene expression, transgenic

## Abstract

Drought is a primary environmental factor limiting maize production globally. Although transferring a single gene to maize can enhance drought resistance, maize response to water deficit requires further improvement to accommodate the steadily intensifying drought events worldwide. Here, we generated dual transgene lines simultaneously overexpressing two drought-resistant genes, *ZmVPP1* (encoding a vacuolar-type H+ pyrophosphatase) and *ZmNAC111* (encoding a NAM, ATAF, and CUC (NAC)-type transcription factor). Following drought stress, survival rates of the pyramided transgenic seedlings reached 62–66%, while wild-type and single transgene seedling survival rates were 23% and 37–42%, respectively. Maize seedlings co-expressing *ZmVPP1* and *ZmNAC111* exhibited higher photosynthesis rates, antioxidant enzyme activities, and root-shoot ratios than the wild type, and anthesis-silking intervals were shorter while grain yields were higher under water deficit conditions in field trials. Additionally, RNA-sequencing analysis confirmed that photosynthesis and stress-related metabolic processes were stimulated in the dual transgene plants under drought conditions. The findings in this work illustrate how high co-expression of different drought-related genes can reinforce drought resistance over that of individual transgene lines, providing a path for developing arid climate-adapted elite maize varieties.

## 1. Introduction

Drought is a common environmental factor that often severely impacts maize growth and yield. Thus, the development of drought-resistant varieties can help to ensure that maize is produced in sufficient quantities for food, livestock feed, and fuel despite stochastic drought events and steadily increasing consumer demand. In particular, increasing maize yields under drought stress can be especially challenging due to the low heritability of stress resistance-related traits [1,2]. Drought resistance is well-understood to be a polygenic trait regulated by numerous genes and other factors [3]. In grain crop cultivation, the degree of damage caused by water deficit depends on soil water potential gradients, and can differ among plant species and developmental stages [4,5]. Studies of drought resistance in maize are largely focused on seedling survival rates after drought stress, yield-associated seed traits, root traits, and photosynthetic efficiency, among others [6,7,8,9]. Since the early 21st century, drought events occur more frequently from spring to summer in Northern China, a major maize growing region [10]. Consequently, water deficit threatens germination and seedling growth in the spring. Therefore, the seedling stage is very important in maize research, and events in the seedling stage have a direct bearing on later developmental stages, and potentially impact yield.

Due to the complexity of drought-related traits, classical long-term, empirical selection breeding strategies typically fail to stabilize yield in drought-resistant varieties [11]. Marker-assisted selection, transgenic, or genome-edited lines have been developed to simultaneously increase yield and abiotic stress resistance [12,13,14]. However, deploying these genes or variants to improve drought resistance in commercial lines remains challenging for breeders. Several drought-related genes have already been identified in maize seedlings, such as *ZmNAC111*, *ZmVPP1*, *ZmPP2C-A10*, *ZmPP2C26*, *ZmGRXCC14*, *ZmTIP1*, *ZmSRO1d*, etc. [3,15,16,17,18,19,20,21]. Among these genes, *ZmNAC111* harbors a promoter region miniature inverted-repeat transposable element (MITE) that serves as a recognition site for RNA-directed DNA methylation and H3K9 dimethylation, resulting in its transcriptional repression. Under well-watered conditions, *ZmNAC111* overexpression has no obvious phenotypic effects on morphology or other agronomic traits compared with wild-type (WT) A188. However, under water-deficit conditions, transgenic overexpression of *ZmNAC111* confers drought resistance through up-regulation of drought-responsive genes, leading to enhanced water use efficiency (WUE) and increased seedling survival rates [15]. Through a different transcriptional regulatory mechanism, a 366-bp insertion in the *ZmVPP1* promoter region increases its expression in drought-resistant maize lines, with *ZmVPP1* overexpression leading to enhanced water-use efficiency and root development, further improving resistance to drought in maize seedlings. Importantly, both transgenic lines have higher grain yields under water stress conditions than WT, most likely due to a shorter anthesis-silking interval (ASI) [16]. ZmPP2C-A10 participates in abscisic acid signaling and can suppress drought resistance [17]. The *ZmPP2C26* gene was used to generate the *ZmPP2C26L* and *ZmPP2C26S* (carrying a 213 bp deletion) isoforms, which negatively regulate drought resistance in maize through dephosphorylation of ZmMAPK3 and ZmMAPK7 [18]. Transgenic *ZmTIP1* expression enhances drought resistance by activating longer growth of root hairs [19], whereas transgenic *ZmSRO1d* leads to higher seedling survival rates and grain yield in maize by neutralizing stomatal reactive oxygen species (ROS) [20]. Thus, individual drought-related genes such as these can each partially contribute to improving seedling stage drought resistance in maize.

Since the drought-resistant phenotype is regulated by many genes, improving this trait may be more effectively achieved by pyramiding multiple genes in a single variety [22,23]. For instance, stacking *TsVP* and *BetA* in maize results in stimulating vegetative growth and increasing yields to a greater degree than either transgene alone [22]. Similarly, pyramiding *HVA1* with *mtlD* can enhance survival rate and biomass production during drought stress compared with the effects of either individual transgene [23]. In the light of other studies that showed stacking genes could further enhance the drought resistance conferred by individual genes, we examined whether stacking *ZmVPP1* with *ZmNAC111* could further increase the effects of increased seedling survival, upregulated drought responsive gene expression, root development, or yield under drought conditions observed in transgenic A188 plants expressing either gene alone. We hypothesized that pyramiding these genes could enhance both the expression of drought-responsive genes and stimulate root development under limited water availability, resulting in a high-yield, drought-resistant line. In this study, we generated dual transgene maize plants by crossing *ZmVPP1* and *ZmNAC111* overexpression lines then examined drought resistance of the gene-stacked plants for comparison with that of single transgene and wild-type (WT) plants. Our data demonstrate that co-expressing *ZmVPP1* and *ZmNAC111* in the drought-sensitive maize line A188 confers drought resistance, improves water use efficiency, increases antioxidant enzyme activity, reduces ASI, and stimulates stress-responsive gene expression under prolonged water deficit conditions.

## 2. Materials and Methods

### 2.1. Plant Materials

*ZmVPP1*-transgenic and *ZmNAC111*-transgenic materials were obtained from previous reports [15,16]. Two transgenic A188 lines, each homozygous for *ZmVPP1* or *ZmNAC111* alleles obtained from the B73 inbred line, were crossed to produce hybrids during the summer of 2016 in our experimental field in Beijing, China (40°08′14″ N, 116°11′52″ E). We planted the hybrid (F1) progeny and harvested seeds (F2) generated through self-pollination during the winter of 2016 in Sanya, China (18°39′10″ N, 109°19′23″ E). Next, we planted approximately 200 F2 seeds and identified double-positive plants for self-pollination by PCR analysis (Appendix A). Each ear of F3 was planted with 3 rows; no less than 30 plants and all double-positive rows were selected for self-pollination. To further verify the homozygous pyramided lines, we continued self-pollinating for two subsequent generations with parallel PCR screening in 2018. Finally, we obtained homozygous pyramided lines (*A3×B7* and *A4×B1*) in the F6 generation and their progeny of self-pollination was used for phenotypic experiments. The WT (inbred line A188), single transgene lines, and pyramided lines were used to investigate drought resistance. Letters indicate the followings transgenic lines: *A3*, *ZmUbi1:ZmVPP1-OE3*; *A4*, *ZmUbi1:ZmVPP1-OE4*; *B1*, *ZmUbi1:ZmNAC111-OE1*; *B7*, *ZmUbi1:ZmNAC111-OE7*; *A3×B7*, *ZmUbi1:ZmVPP1-OE3×ZmUbi1:ZmNAC111-OE7*; *A4×B1*, and *ZmUbi1:ZmVPP1-OE4×ZmUbi1:ZmNAC111-OE1*.

### 2.2. Phenotypic Analyses at Seedling Stage

Planting and drought treatment methods were performed as previously described [16]. Drought resistance was examined at the two-leaf seedling stage in more than 18 plants from each line in each replicate, and the statistical analyses compared phenotypic data from at least three independent experiments. Photosynthetic measurements were collected using the second true leaf of two-leaf stage seedlings with a LiCor-6400 system (LI-COR, Inc., Lincoln, NE, USA), according to the manufacturer’s instructions. Water use efficiency (WUE), the ratio of photosynthesis to transpiration, was used to estimate carbon uptake relative to water loss. We performed three independent tests with no less than 18 seedlings per treatment for each genotype. After the seeds were sterilized and germinated, they were placed in a nutrient solution for hydroponic cultivation using the methods of Liu et al. [24]. The roots of the hydroponic maize seedlings were then washed, and the fresh weights of the aboveground organs and roots were weighed. Fresh materials were then placed in a constant-temperature drying oven at 65 °C for 72 h before weighing to determine their dry mass. We performed three independent hydroponic trials and tests with no less than 18 seedlings from each genotype. 

For the detection of reactive oxygen species (ROS), the second true leaf was collected from seedlings at the two-leaf stage on day 12 of full irrigation or drought treatment. Leaves were incubated with 100 μg mL^−1^ 3,3′-diaminobenzidine (DAB) or nitro blue tetrazolium (NBT) solution for 16 h at 37 °C. Stained leaf samples were boiled in decolorizing solution (ethanol:glacial acetic acid = 3:1) to remove chlorophyll, then photographed directly using a camera (Canon EOS R10, Tokyo, Japan). Superoxide dismutase (SOD), peroxidase (POD), and ascorbate peroxidase (APX) were measured by spectrophotometry. Briefly, 0.5 g of leaf samples were homogenized in 5 mL chilled extraction buffer containing 0.05 mM phosphate buffer (pH7.8) and 1% polyvinyl pyrrolidone. After centrifugation at 10,000× *g* for 10 min at 4 °C, supernatants were collected for enzyme activity assays. The activities of SOD, POD, and APX were measured at 560 nm, 470 nm, and 290 nm as described previously [25]. Malondialdehyde (MDA) content was measured at 532 nm and corrected for non-specific turbidity by subtracting the absorbance at 600 nm and 450 nm, as described previously [25]. Proline content was measured at 520 nm and calculated according to a standard curve, as described in previous reports [26].

### 2.3. Drought Resistance in the Field 

Plant drought resistance was examined from May to September of 2019 and 2020 in our experimental field (Zhuozhou, China. 39°27′49″ N, 115°51′05″ E). A rain-off shelter (32 × 20 m, length × width) was built, which was open on sunny days to prevent shading, and closed on cloudy days to shield plants from rainfall. Drought treatments were divided into watered plots and drought plots, with three replicates. All materials were sown on 10 May 2019 or 2020, with a planting density of 58,000 plants/hectare, and were harvested on 8 September 2019 or 2020. The watered plot was irrigated throughout the growing season. In contrast, the water supply ceased after sowing for 29 days on drought plots, with the exception of two irrigations at the V7 and R1 stages. The drought plots were irrigated with approximately half the water of the watered plots, and more severe drought treatments were performed than in our previous studies. All harvested ears were naturally dried and weighed.

### 2.4. RNA-Seq Analyses 

The planting and drought treatments were performed as previously described methods [27]. For two-week-old maize seedlings, one duplicate was subjected to drought treatment, while the other duplicate was grown under full irrigation. After being drought treated for 13 days, total RNA was isolated for each sample from the second leaves of six test plants at the three-leaf stage using a TRIzol reagent kit (Biotopped). All samples included three biological replicates. The libraries were constructed and sequenced using the DNBseq platform (paired-end 150-bp reads) at BGI-Wuhan (Wuhan, China). Approximately six gigabases of raw data were obtained for each sample (accession number: PRJNA889460). After removing adapters and low-quality reads by fastp with default parameters [28], the filtered reads were aligned to the maize reference genome (B73_RefGen_v4) through HISAT2 with default parameters. Mapped reads with a quality score of >30 were grouped based on their position in the genome. featureCounts was used to calculate the abundance (fragments per kilobase of exon per million mapped fragments; FPKM) of each gene [29]. The differentially expressed genes (DEGs) between pyramided lines and WT were identified by DEseq2 with default parameters [30]. GO ontology was performed using AgriGO [31]. Drought-responsive gene expression was analyzed by qPCR (Quantitative real-time Polymerase Chain Reaction) on an ABI StepOnePlus system with SYBR Premix reagents (Takara Bio, Kyoto, Japan). cDNAs were obtained from total RNA (2 μg) using a reverse transcriptase kit (Promega, Madison, WI, USA). Relative expression was calculated using the CT (2^−∆∆CT^) method and normalized with *ZmUbiquitin1* as an internal reference.

## 3. Results

### 3.1. Overexpression of ZmVPP1 and ZmNAC111 Together Increases Seedling Survival Rate under Drought Conditions

In order to improve drought resistance in maize, we examined previously established four independent transgenic lines, two overexpressing *ZmVPP1* (lines *A3* and *A4*) and two lines overexpressing *ZmNAC111* (*B1* and *B7*) in maize variety A188 [15,16]. We then sought to determine if the combined overexpression of *ZmVPP1* and *ZmNAC111* could further enhance drought resistance by pyramiding the transgenes through gradual sexual hybridization. After obtaining F1, in order to obtain homozygous pyramided lines, two generations of self-pollination were identified by PCR. Throughout the process of two generations, plant height and inflorescence development in the pyramided plants were consistent with the wild-type (WT) phenotype. Following confirmation that both transgenes were indeed present in two independent pyramided lines (*A3×B7* and *A4×B1*), drought treatments were applied to 2-leaf seedlings to evaluate the pyramided effects of *ZmVPP1* and *ZmNAC111* overexpression on seedling survival under drought conditions (Figure 1). Based on the significantly higher survival rate conferred by each transgene, individually, the pyramided plants were administered relatively severe drought treatments by withholding water for 35 days with a 3-day rewatering period prior to the assessment of seedling survival. The survival rate (SR) of plants harboring a single transgene ranged from 37 to 42%. Plants carrying both transgenes had SR values ranging from 62 to 66% (Figure 1A,B). By contrast, wild-type A188 maize plants exhibited a 23% survival rate. The combined overexpression of *ZmVPP1* with *ZmNAC111* could indeed increase maize seedling survival rate under drought stress over that of WT or plants expressing either gene alone (Figure 1C,D).

### 3.2. Photosynthesis Rate, Water Use Efficiency, and Antioxidant Enzyme Activity Are Elevated in Gene Pyramided Seedling Leaves under Drought Treatment

Based on the respective roles of *ZmVPP1* and *ZmNAC111* in photosynthesis rate, stomatal conductance, and transpiration rate, which are all reduced under water deficit in both single transgene and WT plants [15,16], we next compared the photosynthetic capacity of pyramided plants with that of WT seedlings over 12 days of drought treatment (Figure 2A–D). During the first three days of drought treatment, seedlings of all maize lines displayed photosynthesis rates similar to WT. However, photosynthetic indices, except WUE, all significantly declined by day 6 of drought treatment. At day 9 of drought treatment, photosynthetic rates were higher in seedlings expressing both transgenes compared to WT, while at day 12, pyramided plants both exhibited markedly higher photosynthetic rates and water use efficiency (i.e., the photosynthetic rate relative to the transpiration rate) than WT. These results indicated that pyramiding *ZmVPP1* and *ZmNAC111* in maize could improve seedling resistance to drought stress. 

Since oxidative damage from reactive oxygen species (ROS) accumulation plays a well-established role in drought stress, we quantified oxidative damage, MDA content, and proline accumulation in drought-resistant transgenic and WT plants. At day 12 of drought treatment, pyramided plants had significantly higher water use efficiency than WT, so leaf tissues were collected at this time point for ROS accumulation and antioxidant activity assays. Histochemical detection of H_2_O_2_ and O^2−^ by DAB and NBT staining, respectively, showed no difference in ROS staining between WT and transgenic lines under well-watered conditions. By contrast, H_2_O_2_ levels were obviously lower in transgenic plants than in WT on day 12 of drought treatment (Figure 2E). Similarly, O^2−^ staining was also lower in transgenic lines than WT on day 12 of drought conditions (Figure 2F). In light of their roles in eliminating ROS, we also quantified the enzymatic activities of SOD, POD, and APX. We found that the activities of all three enzymes were significantly higher in pyramided seedlings at day 12 of drought treatment compared to that in WT plants, while no differences were detected among WT and transgenic lines under well-watered conditions (Figure 2G–I). After 12 days of drought treatment, we found that MDA content was lower in the transgenic lines than in WT, with pyramided plants exhibiting a 98~134% increase from well-watered conditions compared to a 255% increase in WT (Figure 2J). Quantification of the osmoprotectant, proline, showed that pyramided plants accumulated higher proline levels than WT, with a 53~59% increase from well-watered conditions in transgenic seedlings versus a 38% increase in WT (Figure 2K). The lower ROS and MDA accumulation, along with increased antioxidant activity and proline contents in the transgenic lines likely contributed to their enhanced resistance to drought stress.

### 3.3. Gene Pyramiding Enhances Root Development and Root-Shoot Ratio

Since the ability to acquire water from greater depths in soil when surface moisture is depleted is essential for drought resistance in maize [32], we next examined the dry weight of shoots and roots of each line. No obvious impacts on growth were observed in either the aboveground organs or roots of healthy transgenic seedlings (Figure 3A). In addition, the average root dry weight of *A3×B7* and *A4×B1* stacked lines was greater than that of WT (Figure 3B, left), while the aboveground dry weight between the pyramided and WT seedlings is not significantly different (Figure 3B, middle). Notably, pyramided lines had slightly higher root-to-shoot ratios than WT (Figure 3B, right), which may have contributed to the enhanced seedling survival under drought stress.

### 3.4. ZmVPP1 and ZmNAC111 Overexpression Leads to Shorter Anthesis-Silking Interval and Higher Yield during Water Deficit in the Field 

Since maize is monoecious, with male and female inflorescences emerging at different locations on a single plant, water deficit is well-known to result in the asynchronization of tassel and ear development. We next examined plant phenotypes and anthesis-silking interval (ASI), which is related to pollination efficiency, under drought conditions for two years in 2019 and 2020 in a rain-off shelter of Zhuozhou experimental field, China (Figure 4 in 2019 and Appendix A in 2020). The two transgenic lines and WT were simultaneously planted under rain shelters in irrigated and non-irrigated fields. Under full irrigation, no difference among lines was detectable in plant height, ASI, or yield traits (Figure 4A–C). By contrast, under water deficit conditions, plant height decreased by 4.40–12.17 cm in all maize lines (Figure 4D). Notably, under water deficit, grain yield in both pyramided plants was greater than that of WT plants, and ASI was significantly shorter (2.96–3.42 days versus 5.17 days) in WT (Figure 4E,F). The phenotypic results of different genotypes in 2020 (Appendix A) are consistent with those in 2019 (Figure 4). These collective results indicated that stacking *ZmVPP1* and *ZmNAC111* could enhance maize drought resistance to improve yields during water deficit in the field. 

### 3.5. Abiotic Stress-Related Genes Are Activated in Pyramided Maize Plants under Drought Conditions

To investigate whether the combined overexpression of *ZmVPP1* and *ZmNAC111* was indeed responsible for the observed increase in maize drought resistance, we conducted a whole transcriptome analysis of both pyramided and WT lines under full irrigation and drought conditions. To this end, we collected second leaves for each sample from six seedlings (three-leaf stage) of each pyramided line and WT grown in the greenhouse at day 13 of drought or full irrigation treatments for RNA-sequencing analysis. After trimming reads and filtering for quality and alignment, we identified 2500 DEGs (Figure 5) between the pyramided lines and WT (|fold change| > 2 and *p* < 0.05). Using k-means clustering, the DEGs were classified into four clusters (Figure 6A), including up-regulated genes (Cluster1, 541 DEGs), down-regulated genes (Cluster 2, 206 DEGs mainly down-regulated in pyramid lines; Cluster 3, 529 DEGs downregulated in all lines), and variable response genes (Cluster 4, 1224 DEGs). Subsequent gene ontology analysis suggested that DEGs in Cluster 1 were enriched for terms associated with abiotic stress response and hormone stimuli (Figure 6B). By contrast, down-regulated DEGs in Cluster 2 showed enrichment for GO terms related to photosynthetic processes, while down-regulated DEGs in Cluster 3 were mainly involved in metabolic processes (Figure 6B). By contrast, no GO terms were significantly enriched in Cluster 4. We noted that numerous transcription factor genes were up-regulated among the DEGs in Cluster 1, including the well-studied AP2/ERF, bZIP, IAA, and bHLH drought-related transcription factor families [33,34,35]. To confirm the RNA-seq results, qPCR-based relative expression analysis was conducted for *ERF7*, *bZIP41*, *IAA24*, and *bHLH132*, which were all significantly up-regulated in the pyramided lines and WT after drought treatment (Figure 6C). These findings cumulatively suggested that the combined overexpression of *ZmVPP1* and *ZmNAC111* can enhance maize response to drought conditions.

## 4. Discussion

Drought-induced abiotic stress poses an ongoing, major challenge for the production of maize and other staple grains in growing regions globally. Therefore, drought resistance mechanisms and the recovery capacity after drought treatment in crops serve as a long-term focus of intense agronomic research interest [36]. Adaptation to drought conditions enables plants to persist, if not thrive, during water deficit and rapidly restore their normal physiological functions after rehydration. Previous and ongoing research has established that drought resistance is a quantitative trait determined by a highly complex gene network. Thus the gene-pyramiding strategy is promising to achieve a greater effect than manipulating a signal gene. Recent studies have shown that increasing *ZmVPP1* or *ZmNAC111* expression can enhance the response to water deficit in maize [15,16]. In the current study, we crossed the transgenic maize lines harboring *ZmVPP1* or *ZmNAC111*, individually, to stack these drought-resistant genes and potentially enhance drought resistance over that of the parental lines. In seedlings, survival rates following drought treatment were higher in two dual-transgenic (*A3×B7* and *A4×B1*) lines than in either parental line (*A3*, *A4*, *B1*, *B7*) or the wild type (Figure 1).

Plants have evolved several complex mechanisms to withstand drought conditions, such as synthesizing stress-related hormones, up-regulating antioxidant enzymes and metabolites, and stimulating root development. These mechanisms help to modulate stomatal conductance, scavenge reactive oxygen species, and obtain water from deeper in soil [37,38,39]. In the current study, stomatal restriction occurred rapidly, in the initial stage (0–3 days) of drought. However, photosynthesis did not slow immediately, with the rate of photosynthesis remaining at physiological levels for three days (Figure 2). From an agronomic perspective, an ideal strategy for enhancing drought resistance is to improve water use efficiency by balancing CO_2_ influx and transpiration. For example, overexpression of *BLINK1* in Arabidopsis, results in stomatal movement depending on light, and can increase plant biomass by 2.2-fold without penalizing carbon fixation processes [40]. In this study, pyramiding these genes also resulted in higher WUE during exposure to severe drought (Figure 2A–D), potentially contributing to the increased survival rates among maize seedlings carrying both transgenes. Our data show that ROS production increases linearly with the duration of water stress. The ROS-scavenging antioxidant enzymes, SOD, POD, and APX, play an essential role in protecting plants against oxidative damage caused by excess ROS accumulation [25]. In this study, antioxidant activity and proline content were significantly higher in the pyramided lines compared to WT at day 12 of drought treatment, while ROS and MDA levels were lower in the transgenic lines (Figure 2E–K). This increased antioxidant activity and higher concentration of proline, as an osmoprotectant, likely contributed to the drought-resistant phenotype of the transgenic maize seedlings.

Plant roots are phenotypically plastic and highly sensitive to drought stress, and thus a temporary water shortage can promote root growth [41]. It is therefore likely that developing a relatively deep and strong root system can facilitate water uptake from deeper in the soil, potentially increasing survival rates at the seedling stage [32]. Overexpression of the root developmental regulators *ZmVPP1*, *ZmTIP1*, or *ZmLRL5* can promote root development, resulting in increased root biomass and root hair elongation, and greater resistance to drought conditions [16,19,42]. In the pyramided lines generated here, both root development and root-shoot ratio were increased over that in WT, which could at least partially explain the observed drought resistance in seedlings (Figure 3), and could enable surviving seedlings to thrive after water was replenished. During drought stress, pollination and kernel development are dependent on the transport of assimilates. Under drought stress, photosynthate assimilates are partitioned into discrete compartments to modulate plant growth. In particular, the metabolite trehalose 6-phosphate take effect on controlling the production and partitioning of assimilates during the reproductive stage of development. Fine-tuning trehalose 6-phosphate production can potentially ensure that carbon needed for grain production is appropriately allocated from source tissues throughout the whole plant during water deficit [43]. Interestingly, we found that pyramiding the *ZmVPP1* and *ZmNAC111* transgenes results in shorter ASI and increased grain yield compared to WT under water deficit in the field (Figure 4 and Appendix A). The findings in this study together establish a framework for molecular breeding-based enhancement of drought resistance in maize.

## 5. Conclusions

In this study, transgenic maize lines co-expressing *ZmVPP1* with *ZmNAC111* were generated. Maize seedlings with pyramided transgenes had higher survival rates and a greater root-shoot ratio than WT, as well as shorter ASI and higher yield under drought treatment in field conditions. RNA-seq analysis also identified a large set of up-regulated drought-responsive genes, which could explain the higher growth and yield of pyramided lines during water deficit. Gene pyramiding is thus a viable approach to further improve drought resistance in maize with drought-responsive genes.

## Figures and Tables

**Figure 1 genes-14-00008-f001:**
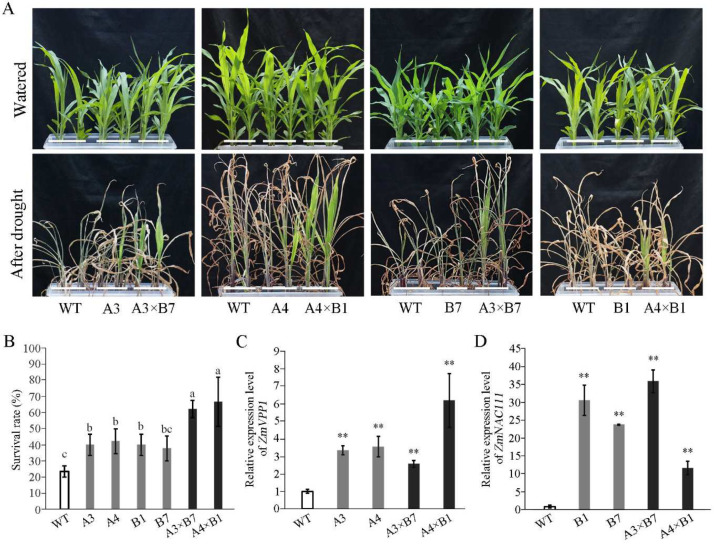
Analysis of drought resistance at seedling stage. (**A**) Representative photographs of maize seedlings before and after drought treatment. (**B**) Statistical analysis of seedling survival rates. 54 seedlings from each line were examined in three independent tests. Different letters indicate significant differences. (*p* < 0.05, one-way ANOVA using Tukey’s multiple comparison tests in SPSS17.0). (**C**,**D**) *ZmVPP1* and *ZmNAC111* transcript levels in transgenic and WT maize. Data are average of three independent tests ± s.d. (** *p* < 0.01; two-sided *t*-test). *A3*, *ZmUbi1:ZmVPP1-OE3*; *A4*, *ZmUbi1:ZmVPP1-OE4*; *B1*, *ZmUbi1:ZmNAC111-OE1*; *B7*, *ZmUbi1:ZmNAC111-OE7*; *A3×B7*, *ZmUbi1:ZmVPP1-OE3×ZmUbi1:ZmNAC111-OE7*; *A4×B1*, *ZmUbi1:ZmVPP1-OE4×ZmUbi1:ZmNAC111-OE1*. The same abbreviations are used in all experiments.

**Figure 2 genes-14-00008-f002:**
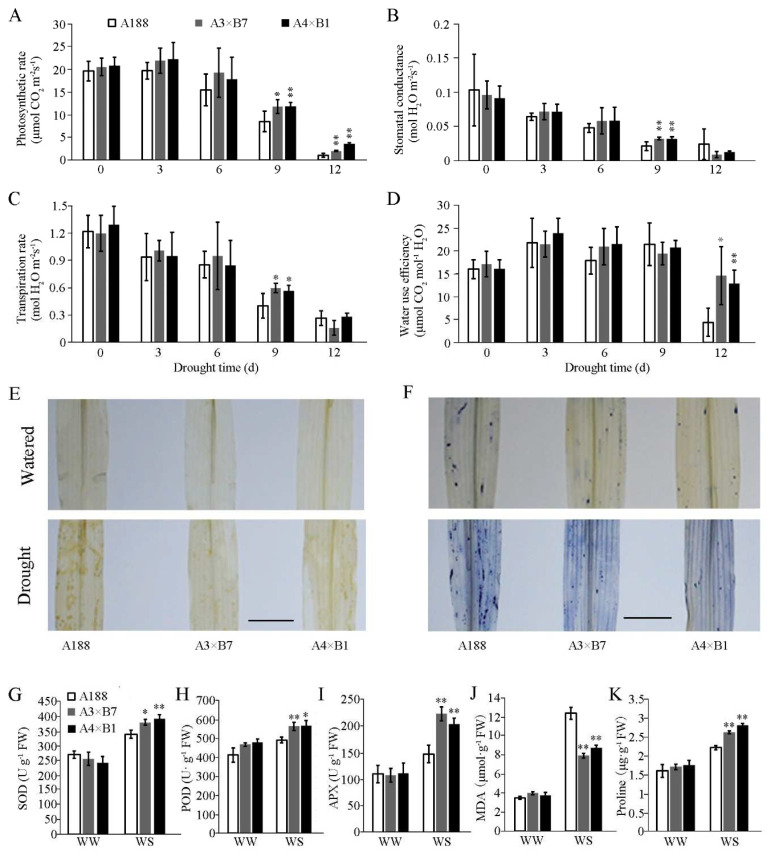
Photosynthesis rates and antioxidant enzyme activities in transgenic and WT maize seedlings in the greenhouse. (**A**) Photosynthesis rate, (**B**) stomatal conductance, (**C**) transpiration rate, and (**D**) water-use efficiency were compared over 12 days of drought stress. (**E**) DAB staining to detect H_2_O_2_ levels in leaves of well-watered or drought-treated transgenic and WT plants at day 12 of treatment. (**F**) NBT staining to detect O^2−^ production in leaves drought-treated or well-watered transgenic and WT plants at day 12 of treatment. The leaf samples were used to detect (**G**) SOD activity, (**H**) POD activity, (**I**) APX activity, (**J**) MDA content, and (**K**) proline content. WW, well-watered; WS, water stress. Bar = 1 cm. Data were obtained from three independent tests with no less than 18 seedlings per genotype. Data are the average of three independent tests ± s.d. (* *p* < 0.05, ** *p* < 0.01; two-sided *t*-test).

**Figure 3 genes-14-00008-f003:**
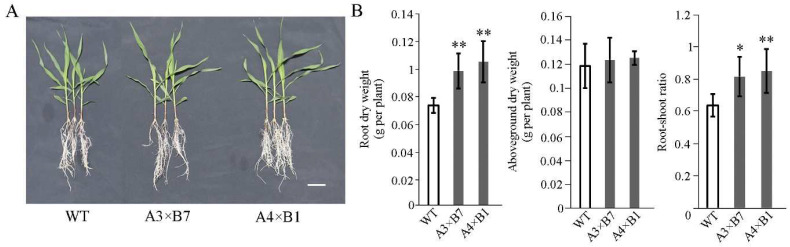
Analysis of root systems in transgenic and WT maize seedlings. (**A**) Representative photographs of WT and transgenic maize root systems. Bar = 5 cm. (**B**) Statistical analysis of root dry weight, aerial tissue dry weight, and root-shoot ratios of WT and transgenic maize under full irrigation (*n* = 6 plants per line). The experiment was conducted three times. Data are average of three independent tests ± s.d. (* *p* < 0.05, ** *p* < 0.01; two-sided *t*-test).

**Figure 4 genes-14-00008-f004:**
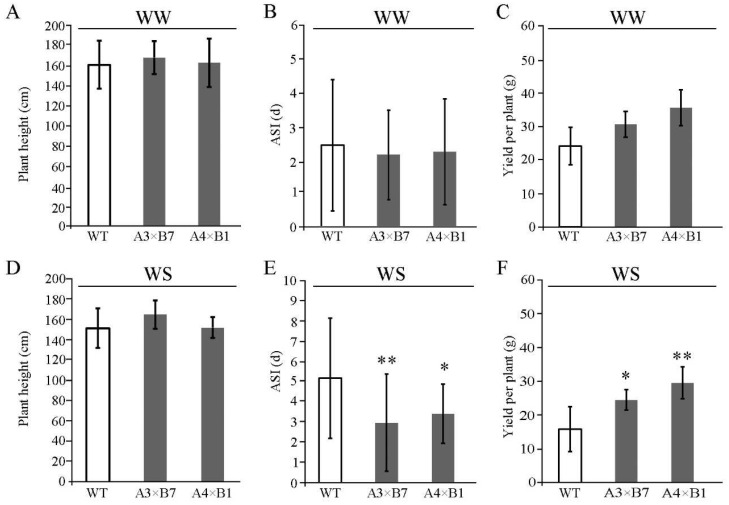
Transgenic and WT maize drought response under field conditions. Statistical analysis of (**A**) plant height, (**B**) ASI, and (**C**) yield per plant for each line under well-watered conditions from May to September 2019 in the field. Statistical analysis of (**D**) plant height, (**E**) ASI, and (**F**) yield per plant for each line under drought conditions from May to September of 2019 in the field. At least 8 plants per line were used in each replicate and independently replicated three times. Data are average of three independent tests ± s.d. (* *p* < 0.05, ** *p* < 0.01; two-sided *t*-test).

**Figure 5 genes-14-00008-f005:**
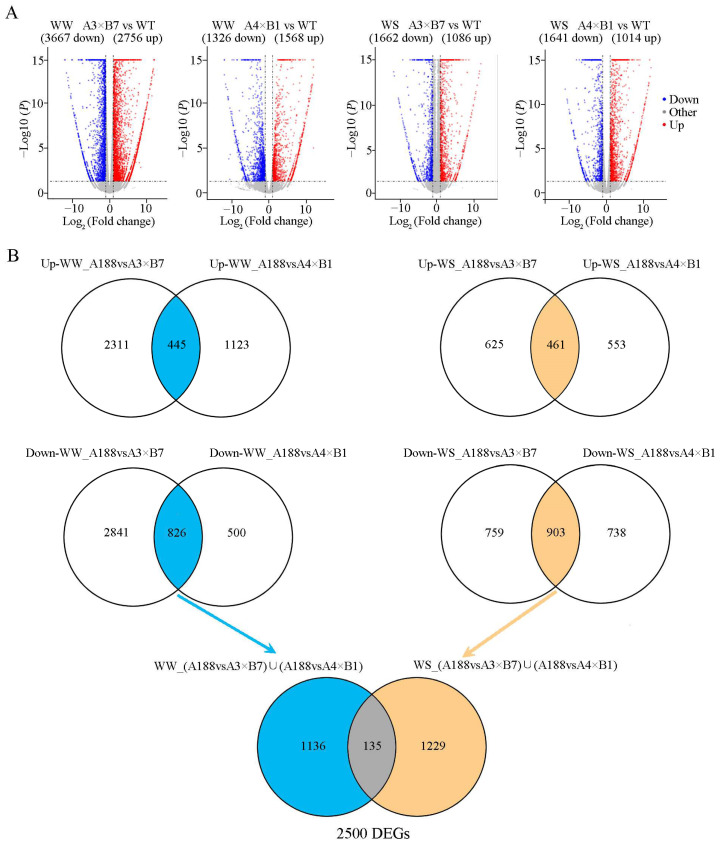
Transcriptomic analysis of dual transgene and WT maize under full irrigation and drought conditions. (**A**) Volcano plots of DEGs (*p*-value < 0.05 and |log2|-fold change > 1). (**B**) Venn diagrams of DEGs for the dual transgene and WT maize under full irrigation and drought conditions.

**Figure 6 genes-14-00008-f006:**
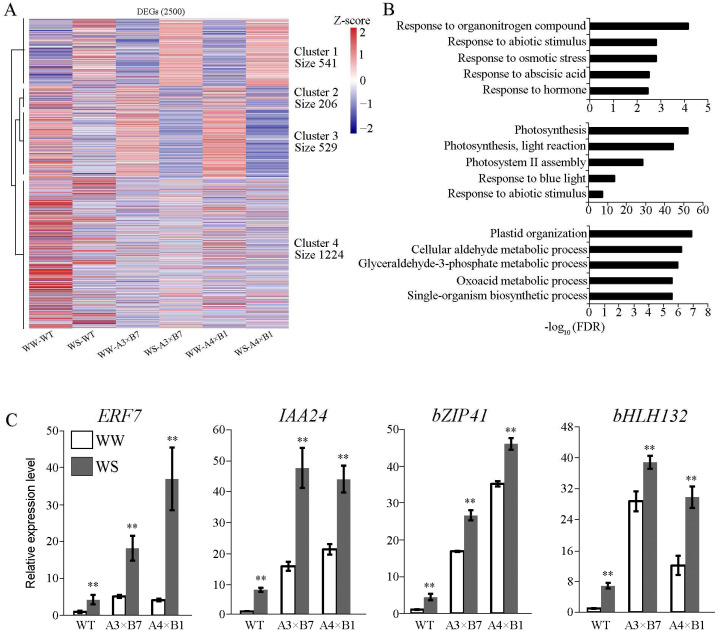
Cluster analysis of DEGs. (**A**) Hierarchical clustering analysis of 2500 DEGs. (**B**) AgriGO Gene Ontology enrichment analysis of biological pathway terms for Clusters 1–3. (**C**) qRT–PCR validation of increased gene expression for drought-responsive transcription factors in dual transgene maize seedlings under full irrigation or drought conditions. Data are average of three independent tests ± s.d. (** *p* < 0.01; two-sided *t*-test).

## Data Availability

In this study, raw data of RNAseq have been deposited in the NCBI under accession number PRJNA889460.

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
