# Peer review of "Co-Expression of *ZmVPP1* with *ZmNAC111* Confers Robust Drought Resistance in Maize"

_genes, 2022, doi:10.3390/genes14010008_

Round 1

Reviewer 1 Report

INTRODUCTION Authors should explain more about the function of selected genes in this study.

Surprisingly nothing has been written about the role of ZmVPP1 and ZmNAC111, and why these were used for pyramiding, etc.

Methods needs to be elaborated. It is not evident how hybrids were produced.

In manuscript there are some grammatical mistakes which should be corrected (In Line no. 136, 305).

In line no. 68, please write the full form of WT.

Mention the full form of both the ZmVPP1 and ZmNAC111 genes.

Drought stress enhances the generation of reactive oxygen species (ROS) and to recover from these species plant increase the antioxidant machinery. So, in your study you can also check the levels of ROS and antioxidants.

The line 281-284 is very long, break it into two lines.

You can also check the drought stress related potential metabolite markers, for instance you can check the concentration of amino acids.

There are several typological and spacing error in the manuscript. Please, read the manuscript again to correct these mistakes.

Reviewer 2 Report

The manuscript titled Co-expression of ZmVPP1 with ZmNAC111 confers robust drought resistance in maize”, investigates the co-expression gene in maize robust drought resistance by using cross-pollination to hybridize transgenic maize lines. The authors generated dual transgene lines simultaneously overexpressing both genes. Following drought stress, survival rates of the pyramided transgenic seedlings reached 62%-66%, while wild-type and single transgene seedling survival rates were 23% and 37%-42%, respectively. Maize seedlings co-expressing ZmVPP1 and ZmNAC111 exhibited higher rates of photosynthesis and root-shoot ratios than the wild type, and in field trials, anthesis-silking intervals were shorter while grain yields were higher under water deficit conditions. Additionally, RNA-sequencing analysis confirmed that photosynthesis and stress-related metabolic processes were stimulated in the dual transgene plants under drought conditions and provided a straightforward path for developing arid climate-adapted elite maize varieties.

The manuscript is well-written and in a nice flow, however, I have some considerations before publishing:

1, For Fig. 2, why only WT and A3B7 or A4B1, should have A3 orA4 and B1 or B7 alone, Same with all rest of the figures.

2, Hybrid is complicated. How can it be separate from heterosis even they are in the same background?

Minor revision:

Line: 49, as ZmPP2C-A10 here, and ZmPP2C26 has reported also (PMID: 35463404)

Line 52: “Alternatively” seems weird here.

Line 60: “Is regulated” might be better.

Line 67-68: a one-sentence summary of overall findings should be included here.

Line 71-79: the letter code can be A3×B7 should better understand the cross between two lines.

Line148-150: the sentence is too long and hard to read; please try to break it into two sentences.

Line151: Figure 1A (after drought) should have a dash to mark clearly the same as the watered figure in the upper panel.

Line 177: figure 2 should be labeled as A-D as each figure is different, also please indicated the legend as detailed as possible. Same with line 221 Figure 4.

Line 214: “during water deficit front is different here.

Line259: significance should be indicated in Figure 6C

Line 360: ref.9, journal name is different

Round 2

Reviewer 1 Report

Authors have addressed all the concerns.